# Dysregulation of Mitochondrial Homeostasis in Cardiovascular Diseases

**DOI:** 10.3390/ph18010112

**Published:** 2025-01-16

**Authors:** Ricky Patil, Hui Wang, Matthew Kazaleh, Gorav Ailawadi, Morgan Salmon

**Affiliations:** 1Department of Cardiac Surgery, Michigan Medicine, University of Michigan, Ann Arbor, MI 48109, USA; hhhwang@umich.edu (H.W.); mkazaleh@med.umich.edu (M.K.); ailawadi@med.umich.edu (G.A.); msalmon@med.umich.edu (M.S.); 2Frankel Cardiovascular Center, School of Medicine, University of Michigan, Ann Arbor, MI 48109, USA

**Keywords:** mitochondria, aortic aneurysm, murine model, elastase, cardiac surgery, aorta, inflammation, B-aminopropionitrile

## Abstract

Mitochondria dysfunction plays a central role in the development of vascular diseases as oxidative stress promotes alterations in mitochondrial morphology and function that contribute to disease progression. Redox imbalances can affect normal cellular processes including mitochondrial biogenesis, electrochemical equilibrium, and the regulation of mitochondrial DNA. In this review, we will discuss these imbalances and, in particular, the potential role of mitochondrial fusion, fission, biogenesis, and mitophagy in the context of vascular diseases and how the dysregulation of normal function might contribute to disease progression. We will also discuss potential implications of targeting mitochondrial regulation as therapeutic targets to treat vascular disease formation.

## 1. Introduction

Cardiovascular disease (CVD) is the leading cause of death and a major cause of disability worldwide. Total CVD case prevalence nearly doubled from 271 million in 1990 to 523 million in 2019. Additionally, the number of CVD-related deaths increased from 12.1 million in 1990 to 18.6 million in 2019 [1]. Hence, the search for novel therapies is vital to improving care. In recent years, the relationship between mitochondrial homeostasis and cardiovascular disease has become a growing area of study for potential therapeutic intervention. Mitochondrial dysfunction and resultant oxidative stress enact a crucial role in CVD pathogenesis. Mitochondrial membrane permeability, mitochondrial DNA regulation, mitochondrial fusion and fission, and mitophagy all affect cardiovascular disease pathogenesis. Hence, a better understanding of these mechanisms may contribute to improved medical care. This review aims to discuss recent advances in the connection between mitochondrial health, oxidative stress, and vessel dysfunction.

## 2. Oxidative Stress in Mitochondria

Mitochondria produce the majority of cellular oxidant stress due to their central role in respiration and ATP production. Sources include superoxide radicals from the electron transport chain as well as outer mitochondrial membrane enzymes such as monoamine oxidase and aldehyde oxidase [2,3]. Mitochondrial reactive oxidative species (ROS) have historically been show to play a physiologic role in maintaining cellular homeostasis in processes such as signal transduction and protein modification [4]. Elevated levels of ROS prepare cells for the insult presented by atherosclerosis and other cardiovascular pathologies [5]. Meanwhile, antioxidant molecules such as glutathione, carotenoids, and coenzyme Q10 mitigate the negative effects ROS activity and prevent further harm [6]. However, when the body’s innate protective mechanisms fail, elevated oxidative stress can promote CVD.

### Cellular Effects of Oxidative Stress in Cardiovascular Disease

In atherosclerosis, mitochondrial oxidant stress contributes to tissue damage and plaque expansion. An important mechanism in this process is lipid oxidation. ROS oxidize lipids accumulating in the arterial wall, forming oxidized low-density lipoprotein or ox-LDL, which exacerbates vascular smooth muscle damage. Ox-LDL activates mitochondrial cytochrome C release, supporting VSMC apoptosis. Additionally, ox-LDL damages mitochondrial DNA (mtDNA), resulting in reduced aerobic respiration and reduced complex I expression in vascular smooth muscle cells; this process supports necrotic core and fibrous cap growth in atherosclerotic plaques. Human atherosclerotic plaques demonstrate a lower mtDNA copy number, lower Complex 1 expression, and lower O_2_ consumption rates in the fibrous cap and necrotic core areas. This phenomenon was also replicated in Apolipoprotein E-deficient (*ApoE*^(−/−)^ mouse VSMC [7]. When mouse VSMC mtDNA replication was increased through mitochondrial helicase overexpression, there was an increase in mitochondrial complex abundance, a reduction in ROS production, and a decrease in plaque fibrous cap area and necrotic core size [7]. Hence, in atherogenesis, VSMC oxidant stress affects mtDNA expression, halting the production of vital organelle mechanisms, which results in worsening atherosclerotic damage.

Cardiovascular inflammation also promotes oxidative damage in endothelial cells. In the vessel wall, the endothelial layer serves many functions. Ox-LDL formation induces endothelial dysfunction by enhancing the expression of I-CAM, V-CAM, and P-selectin, resulting in increased adhesion of inflammatory cells [8]. Furthermore, mitochondrial oxidative stress also affects nitric oxide-induced vasodilation, mainly through the ROS-mediated degradation of endothelial NO synthase (eNOS). eNOS degradation upregulates mitochondrial arginase II [6], which reduces arginine, an NO-precursor. This phenomenon is uniquely demonstrated in a separate genetic syndrome known as MELAS (mitochondrial encephalopathy, lactic acidosis, and stroke-like syndrome), where genetic mutations result in impaired mitochondrial electron transport chain production, attenuated energy production, and elevated ROS production [9]. Afflicted patients demonstrate arginine deficiency, resulting in the vasoconstriction of smaller vessels and resultant ischemia. Arginine supplementation in MELAS helps reverse vaso-constrictive episodes, indicating its therapeutic potential [9]. Overall, the mitochondrial oxidative stress witnessed in MELAs can affect endothelial barrier function and vaso-activity, thus providing a new mechanism for future study.

## 3. Mitochondrial Membrane Dynamics

Mitochondria trigger regulated cell death (RCD) through alterations in the inner mitochondrial membrane (IMM) and outer mitochondrial membrane (OMM). These changes redistribute mitochondrial proteins into the cytoplasm, triggering cell death cascades. The two main mechanisms described involve MOMP (outer mitochondrial membrane permeabilization) and MPT (mitochondria permeability transition). MOMP results from pore formation from proteins Bax and Bak, members of the B-cell lymphoma (Bcl-2) family, which oligomerize into rings and arc-shaped structures to create large conductance pores in the OMM. This process then leads to the release of pro apoptotic effectors, namely cytochrome C (CytC), and the activation of the caspase cascade and apoptosome formation [10]. The MPT relies on the selective release of calcium (Ca^2+^) from the ER (endoplasmic reticulum) into the mitochondria. Mitochondrial exposure to pathologic levels of Ca^2+^ leads to the formation of the mitochondrial permeability transition pore complex (PTPC), a voltage-gated high-conductance channel at the interface between the IMM and the OMM. The opening of the PTPC leads to the redistribution of small solutes, loss of mitochondrial membrane potential, and organelle swelling [11,12]. These phenomena play an important role in many cardiovascular processes.

### 3.1. Mitochondria Permeability Transition (MPT) in Ischemia Reperfusion Injury

The role of MPT in ischemia reperfusion injury has been well characterized. Ischemic conditions promote a hypoxic cellular state, reducing electron transport chain (ETC) activity and depressing oxidative phosphorylation. These changes result in electron accumulation among different respiratory complexes and ROS production [13]. Increased ROS, pathologic intracellular Ca^2+^, and decreased mitochondrial membrane potential (∆ψm) induce MPT formation [13,14]. However, MPT formation in the ischemic phase is negatively regulated by ADP and proton accumulation in hypoxic environments, making the relative ADP/ATP during ischemia a determinant of MPT and PTPC formation. During the reperfusion phase, the inhibitory effects of protons and ADP on PTPC formation are lost as the restoration of pO2 activates ATP synthesis and reduces cellular acidosis. Additionally, reperfusion stimulates more ROS production through effectors such as succinate, which stimulates reverse electron transport through Complex 1 of the ETC [15,16]. Other enzymes outside of the mitochondria including xanthine oxidase, NADPH oxidase, and NO synthase also induce ROS production. This positive feedback loop makes the mitochondria the largest source of ROS during reperfusion injury [17]. Hence, pharmacologic protection against reperfusion injury has largely targeted the PTPC mechanism through the administration of cyclosporin A (CsA) in the treatment of ischemic processes [18,19]. CsA, a PTPC inhibitor, has been studied in large-scale prospective trials for the treatment of coronary reperfusion injury, but failed to show large-scale therapeutic benefits. The 2015 CIRCUS trial was a multicenter, double-blinded, and randomized trial where the effect of a pre-PCI bolus injection of cyclosporine (2.5 mg/kg) was tested against regular PCI controls. At the one-year follow-up, cyclosporine administration did not show a difference in the study’s primary outcomes of death, worsening heart failure, heart failure rehospitalization, or adverse left ventricular remodeling [20]. The 2016 CYCLE trial utilized a similar protocol on patients with large infarct STEMI but was also unable to show a difference in time until ST-segment resolution, troponin levels, or level of LV remodeling [21]. It is speculated that the improved prognosis of MI with early revascularization therapy and better medical management makes it more difficult to demonstrate the clinical significance of cardioprotective medications like CsA, regardless of the medication’s biological effect. While CsA’s effect on PTPC and ischemia reperfusion had limited effects in large-scale trials, CsA could still be applicable to other disease processes.

### 3.2. Mitochondrial Permeability Transition and Post Cardiac Arrest Syndrome

The physiological benefits of closing the PTPC have been demonstrated in animal models of post cardiac arrest syndrome (PCAS). After ROSC is achieved, PCAS can cause systemic inflammation, brain damage, myocardial dysfunction, and multi-organ failure [22]. In a porcine model of PCAS, animals were subjected to intermittent bilateral limb ischemia, which reduced post-arrest inflammation and end organ dysfunction. In this experiment, intermittent limb ischemia pre-conditions cells to become ischemic, thus lessening their acute response, specifically, PTPC pore opening. However, when animals were pre-treated with atractyloside, a PTPC opener, the benefits of pre-conditioning were abolished. This result suggests that attenuation of PTPC opening by intermittent ischemic conditioning can attenuate systemic oxidative stress in PCAS [22].

### 3.3. Mitochondrial Permeability Transition and COVID-19 Vasculopathy

PTPC opening also contributes to the endothelial cell dysfunction seen in COVID-19 patients. Human endothelial cells (ECs) were incubated in 10% serum samples from COVID-19 survivors (*n* = 20) or non-survivors (*n* = 10) for 24 h. Mitochondrial potential, calcium accumulation, and ROS production were assessed. ECs incubated in non-survivor serum demonstrated lower mitochondrial potential, a 2.4× shorter time until PTPC opening, and elevated mitochondrial ROS production compared to ECs incubated in survivor serum. The administration of CsA increased mitochondrial potential and lengthened the time until PTPC opening in COVID-19 non-survivor cells alone. These results suggest that PTPC opening and oxidative stress in endothelial cells may contribute to COVID-19-attributed vasculopathy [23]. These findings and others suggest that oxidative stress leads to tissue damage through alterations in mitochondrial membrane dynamics. The PTPC is an important mechanism in this process and may be a target for future medical treatment therapies.

## 4. Mitochondrial DNA (mtDNA)-Driven Inflammation

Mitochondria, hypothesized to have evolved from saprophytic bacteria, can induce a similar immune response to their micro bacterial counterparts [24]. This process occurs in part because of structural similarities between bacterial and mitochondrial DNA (mtDNA). For example, both mtDNA and bacterial DNA have unmethylated CpG island sequences in their structure [25]. Therefore, factors that recognize bacterial DNA like the Toll-Like Receptor 9 (TLR-9) can also recognize mtDNA and elevate downstream macrophage infiltration and p38 MAPK activation [26] (Figure 1). The effects of mtDNA on TLR-9 activity were studied in cardiomyocyte-specific DNase-deficient mice subjected to surgical temporary aortic constriction (TAC). D-Nase deficiency, which upregulates mtDNA levels, led to fulminant myocarditis and dilated cardiomyopathy [27]. The animals demonstrated increased mtDNA deposits in autolysosomes and increased inflammation. However, when TLR-9 activity was inhibited by exogenous inhibitory oligodeoxynucleotides, these detrimental effects were abrogated [27]. The interplay between mtDNA and TLR-9 also affects peripheral vascular performance in systemic inflammatory syndromes. In septic shock and acute respiratory distress syndrome, mtDNA-TLR-9-dependent mechanisms increase endothelial cell (EC) permeability for neutrophils (PMN) and elevated PMN-EC adherence [28]. Overall, by being recognized as a foreign entity, mtDNA triggers effectors like TLR-9 to increase inflammation, making it a key contributor to cardiovascular disease progression.

Another mechanism of mtDNA-induced inflammation is the NOD-, LRR-, and pyrin-domain-containing protein 3 (NLRP3) inflammasome. In autophagy-impaired cardiovascular disease states, the combination of NLRP3 inflammasome and mitochondrial ROS can elevate cytosolic mtDNA leakage, thereby exacerbating cytokine release. For example, in an LPS-induced sepsis model, mice deficient in autophagy protein LC3B and beclin 1 were administered exogenous ATP. The exogenous ATP produced rapid mitochondrial swelling and ROS production, leading to elevated cytosolic mtDNA release [29]. Cytosolic mtDNA contributed to increased NLRP3 inflammasome activation, caspase 1 cleavage, and IL-1B and IL-18 release, which made test subjects more susceptible to LPS-induced mortality. This study links mitochondrial quality control and NLRP3-dependent caspase 1 activation in systemic inflammation. Additionally, nuclear DNA did not translocate to the cytoplasm in this model, making this process unique to mtDNA substrates [29]. These studies suggest that cardiovascular diseases function in part through the promotion of cytosolic mtDNA accumulation and in part through the NLRP3 inflammasome to further promote inflammation.

Another mechanism by which mtDNA contributes to cardiovascular inflammation is the cGAS-STING pathway. The STING protein works with cGAS (cyclic GMP-AMP synthase), a nucleic acid pattern recognition receptor, which recognizes foreign double-stranded DNA (dsDNA). After binding dsDNA, cGAS catalyzes the synthesis of cGAMP or 2′3′-cyclic guanosine monophosphate, which binds STING to trigger downstream effectors such as TBK-1 (TANK-binding kinase 1), IRF-3 (interferon gamma regulatory factor 3), and NF-*K*B (nuclear factor-kappa B), causing downstream cytokine production [30,31,32] (Figure 2). Overall, this mechanism relies on the differentiation between self-DNA and non-self-DNA to incite inflammatory processes [33]. In pathologic states, oxidative stress overwhelms the regulators of mtDNA, leading to increased levels of mtDNA fragments and increased cGAS-STING activation. Intra cellular mtDNA-mediated cGAS-STING interactions have evolved structural modifications to enhance enzyme–substrate interplay. TFAM or Mitochondrial Transcription Factor A, a regulator of mtDNA organization and its transcription machinery, supports the activity of cGAS by incorporating U-turns and bends in mtDNA to help nucleate cGAS dimers [34,35]. In stressed states, the regulators of TFAM are diminished, leading to the elevated leakage of mtDNA into the cytosol and further activation of cGAS-STING. Multiple theories exist on how TFAM dysregulation leads to mtDNA leakage into the cytosol, including elevated mitochondrial membrane permeability and voltage-gated ion channel activation. Pro apoptotic pathways such as the stimulation of caspases by cytochrome C may also upregulate mtDNA leakage [33]. Hence, adaptations in mtDNA structure have allowed for increased cGAS-STING activation and downstream inflammation.

Additionally, extracellular mtDNA secretion has also been documented in animal models of cell death. Immune cells such as eosinophils and neutrophils release mtDNA in response to bacterial killing stimuli. Activated platelets also secrete mtDNA to promote leukocyte activation. It is theorized that platelets release mitochondria into the bloodstream both as exocytosed microparticles and as free organelles. This activates phospholipase A2 IIA (sPLA2-IIA), a bacteria-recognizing enzyme, which recognizes the mitochondria’s homologous structure [36]. This leads to organelle lysis and the release of mtDNA into the bloodstream, resulting in systemic leukocyte adhesion. These processes contribute to inflammatory processes in diseases such as septic shock, acute respiratory failure, and auto-immune pathologies like rheumatic disease [36]. Overall, the breakdown in mitochondrial DNA sequestration activates pro-inflammatory mechanisms at multiple levels of the body’s immune response.

### The cGAS-STING Pathway in Cardiovascular Disease

The activation of cGAS-STING by mtDNA and other substrates has manifested clinically in many cardiovascular diseases. For examples, The New England Journal of Medicine published a series of six cases of STING-associated vasculopathy with onset in infancy (SAVI) resulting from constitutive STING activation as a result of a gain-of-function mutation in the TMEM173 gene [37]. Symptoms included signs of systemic inflammation including fever and vasculopathy including violaceous plaques, ulcerative lesions, digit gangrene, and rash. Histology demonstrated leukocytoclastic vasculitis and micro thrombotic angiopathy of small dermal vessels. Furthermore, when comparing immunofluorescence staining of skin biopsy samples between SAVI patients and controls, SAVI patients demonstrated loss of continuous CD31 staining, an endothelial cell marker, demonstrating damage to the endothelial cell layer as a result of the mutation [37]. Hence, SAVI demonstrates that STING activity has direct effects on vessel health and function.

The cGAS-STING pathway has also been studied in myocardial infarction (MI). In MI, mitochondria and nuclear DNA suppression is disrupted, leading to a maladaptive innate immune response. Murine coronary ligation models of MI indicate that the cGAS-STING pathway modulates IRF-3- and type-1-IFN-driven macrophage recruitment responsible for post-MI inflammation [38]. In concurrent gated microscopy of the wild type, MI subjects’ beating hearts demonstrated increased translocation of nuclear and mitochondrial DNA into the cytosol. Mice that were genetically deficient in STING, c-GAS, or IRF-3 all demonstrated a decreased post-ischemic inflammatory insult, improved ventricular contractility, and improved survival in the infarction model [38]. Hence, ischemia dysregulates DNA sequestration and activates the cGAS-STING pathway, which catalyzes the arrival of post-inflammatory milieu [38].

The cGAS-STING pathway’s activation by mtDNA also directly contributes to atherogenesis; atherosclerotic plaques develop at arterial bends, where the endothelium is exposed to elevated oscillating shear stress (OSS). OSS has been found to cause mtDNA leakage in endothelial cells with the subsequent upregulation of the cGAS-STING pathway, leading to endothelial senescence and atherogenesis. STING was found to be upregulated in human atherosclerotic plaques and aortic arch endothelial cells of ApoE knockout mice (*ApoE*^(−/−)^ on a high-fat diet (HFD) [39]. This effect was attenuated with the genetic knockout of STING [39]. Therefore, mtDNA-driven cGAS-STING activation contributes to atherosclerotic disease in part by contributing to atheroma formation and vessel wall damage.

Aortic aneurysm and aortic dissection, both pathologies characterized by smooth muscle cell loss, have also been attributed to the CGAS-STING pathway. STING knockout mice demonstrated reduced aortic enlargement and elastic fiber fragmentation compared to a wild-type control in murine models of acute aortic dissection, which may indicate that the cGAS-STING pathway plays a role in aortic VSMC alterations through TBK and IRF-3. In vitro studies also demonstrate that STING does participate in ROS-mediated cellular stress. Tissue monocytes engulf fragmented mtDNA to activate the cGAS-STING-IRF-3 axis, which elevates MMP9 (mixed metallopeptidase 9) levels, contributing to further ECM degradation. In addition, the therapeutic targeting of TBK-1 phosphorylation of STING at Ser66 has been employed to preserve aortic architecture and reduce the incidence of AAD [40]. These data suggest that the modulation of this pathway may prove useful in treating aortic pathology.

The mitochondrial DNA activation of the cGAS-STING pathway also plays a role in obesity-induced endothelial inflammation and vascular stress. The treatment of human aortic endothelial cells (HAEC) with palmitic acid (PA) resulted in the elevated perinuclear translocation of STING and IRF-3 phosphorylation and the downstream elevation of inflammatory proteins including ICAM-1, an endothelial adhesion molecule [41]. This effect was reversed by the siRNA-mediated knockdown of the STING protein. Additionally, PA-treated HAEC had elevated levels of cytosolic mtDNA compared to controls, suggesting that PA mediates its effect through mitochondrial damage. The treatment of HAEC with mtDNA from PA-treated HAEC also increased ICAM expression, further validating this mechanism [41]. In a murine model of diet-induced morbidity, wild-type (WT) and STING(−/−) mice were exposed to either a 12-week high-fat diet (HFD) or a normal chow diet (NC). HFD WT mice demonstrated elevated ICAM-1 and phosphorylated IRF-3 in epididymal adipose tissue compared to NC WT mice. This effect was attenuated in the HFD Sting(−/−) mice [41]. Furthermore, in vitro studies of PA-treated HAEC have also demonstrated that cytosolic mtDNA’s activation of cGAS-STING can halt angiogenesis. cGAS-STING’s upregulation of IRF-3 activates the production of MST-1 (mammalian Ste2-like kinase), a contributor to the Hippo pathway [42]. The activation of the Hippo pathway inhibits the end effector protein YAP (yes-associated protein), which is involved in endothelial cell proliferation and migration [43]. This impairs angiogenesis and can have a clinical impact on ischemic wound healing in the clinical setting [43]. Overall, these findings suggest that mtDNA-driven STING activation partakes in obesity-mediated vascular inflammation. The effects of cGAS-STING activation are central to many cardiovascular disease processes. The release of mtDNA in diseased environments catalyzes cGAS-STING, thereby linking systemic inflammatory processes to mitochondrial health. Further inquiry into this mechanism may aid in the development of new pharmacologic agents to treat CVD.

## 5. Mitochondrial Dynamics and Cardiovascular Disease

Investigations into the regulation of mitochondrial dynamics have also become an area of increased study in cardiovascular disease therapy. Processes including mitochondrial fission and fusion have potential roles in disease pathogenesis. Mitochondrial fusion involves the merging of both the inner and outer mitochondrial membranes. In mammals, the three main GTPases involved in mitochondrial fusion are Mitofusin (MFN) 1 and 2 of the OMM and Optic Atrophy 1 (OPA-1) of the IMM. Based on structural studies, MFN mediates oligomerization through an extended antiparallel coiled-coil that brings adjacent OMMs within 100 angstroms of one another, allowing outer membrane fusion to occur [44]. OPA-1 binds to lipid membranes that contain negatively charged phospholipids, namely cardiolipins, a signature lipid of the IMM [45]. The mechanism through which OPA-1 physically aids in IMM fusion is not fully understood, but OPA-1 has been demonstrated to deform liposome surfaces and cause the lengthening of lipid tubules [46]. These activities may contribute to inner membrane fusion. Mitochondrial fusion helps promote cellular health by allowing the mixing of mitochondrial enzymes, metabolites, and gene products. The sharing of mtDNA amongst organelles also reduces the impact of mtDNA mutation, especially in the context of cellular aging.

Mitochondrial fission is coordinated by a cytosolic protein, namely Dynamin-related protein or DRP-1, which is responsible for mitochondrial tubule assembly and the mediation of organelle scission. DRP-1 oligomerizes with mitochondrial surface proteins FIS-1, Mff, MiD49, and MiD51 to form higher-order complexes that constrict mitochondria [47]. Other active components in this process are the endoplasmic reticulum, lysosomes, actin, and cytoskeleton binding proteins INF-2 and Spire1C, which promote mitochondrial contraction in preparation for DRP-1-mediated scission [48]. Fission from the middle of the mitochondrion promotes mitochondrial proliferation, while fission at the end of the mitochondria sequesters damaged material for mitochondrial phagocytosis [49]. Hence, mitochondrial fission is vital to multiple processes such as cell division, the recycling of damaged organelle products, and the distribution of mitochondria to areas of the cell requiring energy.

### 5.1. Mitochondrial Dynamics and Vascular Smooth Muscle Cells

While basal levels of mitochondrial fission are necessary for regular organ function, excessive rates of fission have been associated with CVD [50]. For example, mitochondrial fission plays a role in intimal hyperplasia formation, a response to arterial wall injury. Intimal hyperplasia requires PDGF-mediated VSMC migration for neointima formation. Mitochondrial fission is hypothesized to be vital to this process and is controlled by the PDGF-triggered activation of DRP-1 [50]. In murine VSMC with mutant or absent DRP-1, PDGF-stimulated VSMC lamellipodia formation and VSMC migration are greatly reduced, while PDGF-induced ROS formation and mitochondrial energetics are also stunted [50]. Additionally, in vivo murine models of induced arterial injury with mutant DRP-1 expression produce lower levels of ROS and reduced neointima formation [50]. Therefore, PDGF-driven mitochondrial fission is essential to the role of VSMC in neointima formation. In contrast, the downregulation of mitochondrial fusion proteins has been attributed to VSMC switching from a contractile to a synthetic phenotype [51]. When VSMCs are exposed to PDGF, there is a 50% decrease in MFN-2 production, along with a more fragmented mitochondrial network [51]. Finally, when VSMCs switch to a more proliferative phenotype, their mitochondria change their metabolism from glucose-dependent to fatty-acid-dependent oxidation [51]. This metabolic substrate switching may help drive the cell’s synthetic function. However, this is a less efficient means of energy production and may exacerbate oxidative stress.

Studies that manipulate mitochondrial fusion and fission proteins also demonstrate changes in disease phenotype and progression. For example, in a rabbit model of atherosclerosis, MFN-2 overexpression inhibited the formation of oxidized-LDL and VSMC migration [52]. Morphologic analysis also demonstrated that MFN-2 overexpression inhibited atherosclerosis formation by 66% and reduced intima/media thickness by 74.6% [52]. Hence, the modification of MFN-2 activity may affect CVD disease severity. Mitochondrial fusion proteins can also affect disease pathogenesis through non canonical pathways [52]. For example, in isolated rat VSMC, MFN-2 was shown to mediate oxidative-stress-mediated apoptosis in VSMC through the inhibition of the Ras-PI3K-Akt signaling pathway [52]. This effect was independent of mitochondrial fusion, which indicates that mitochondrial fusion proteins can have an effect on vessel health outside their postulated traditional roles.

Mitochondrial fusion and fission may also affect vessel calcification through VSMC phenotype switching. In stressed physiologic states, synthetic VSMC can demonstrate a more osteogenic phenotype, characterized by the release of calcification-promoting extracellular vesicles that have elevated mineralization capacity [53]. Vessel calcification increases arterial stiffness, promotes ECM degradation, and exacerbates both hypertension and atherosclerosis [53]. The DRP-1 fission protein is highly expressed in calcified areas of human arteries [54]. Additionally, in phosphate-mediated calcification models, there was a reduction in mitochondrial respiratory capacity [55]. Phosphate-treated VSMCs also demonstrate increased mitochondrial fission [56]. Likewise, the disruption of mitochondrial structure was noticed in rodent models of adenine-induced aortic calcification [57]. In the pharmaceutic space, agents such as melatonin and quercetin have been utilized to quell fission-driven vessel calcification in experimental models [56,57]. In summary, dysfunctional mitochondria with elevated fragmentation have reduced respiratory ability and can contribute to calcific disease presentations.

Other clinical presentations have also been linked to mitochondrial fission. For example, animal models of diabetes-induced neointimal hyperplasia targeting DRP-1 have been used to study intimal proliferation [58]. Finally, the relation between mitochondrial fission, alteration in VSMC phenotype, and disease has also been explored in ESRD patient fistula patency and diabetic coronary microvascular disease [50,59]. These studies suggest the clinical application of mitochondrial-fission-related therapeutics could be potentially broad in application. A mitochondrial fission therapeutic known as Isoliquiritigenin, a flavonoid fractionated from *Glycrrhiza glabra*, has been shown to reduce intimal hyperplasia in smooth muscle cells from the human aorta, pulmonary artery, and other sources by acting through DRP-1 [60,61]. In addition, GLP agonists may also treat cardiovascular disease through increased mitochondrial fusion [62]. In vitro GLP-1 administration to VSMC led to more mitochondrial surface area, increased IMM potential, and organelle oxidative capacity [62]. GLP-1 administration was shown to increase MFN-2 protein production and increase the PKA-induced activation of DRP-1 [62]. These studies suggest additional implications for GLP agonists in CVD.

### 5.2. Mitochondrial Dynamics and Endothelial Cells

Mitochondrial dynamics also affect endothelial cell activity. Endothelial cells derive most of their energy from glycolysis, making oxidative phosphorylation only a small fraction of their energy production. Hence, endothelial cell mitochondria have more active functions such as cell migration and angiogenesis. For instance, in VEGFA-cultured human umbilical vein endothelial cells, the knockdown of MFN proteins resulted in excessive mitochondrial fracturing, resulting in reduced cell viability and increased apoptosis [63,64,65]. Specifically, reduced MFN-2 levels resulted in reduced ROS production and mitochondrial-ER uncoupling. Reduced MFN-1 levels also reduced VEGF-stimulated Akt-eNOS signaling [65,66]. MFN protein levels also can affect angiogenesis. In vascular-resident endothelial progenitor cells (VR-EPCs), increased mitochondrial fusion activates pyruvate kinase myozyme 2 (PKM-2), elevating glycolysis and stimulating cell migration, hence promoting angiogenesis [67]. These studies suggest that mitochondrial dysfunction can affect endothelial cell viability, vaso-activity, and new vessel formation. Excessive endothelial mitochondrial fission is also seen in high-glucose states, which may contribute to hyperglycemic vascular injury [67]. Mouse coronary endothelial cell (MCEC) cultures exposed to high-glucose environments demonstrated increased levels of DRP-1 and increased mitochondrial fragmentation. The corresponding in vivo animal experiments also showed reduced OPA-1 and elevated DRP-1 in MCEC [67]. Similar alterations in fusion- and fission-related proteins and subsequent pathogenesis have also been seen in murine retinal endothelial cells exposed to high-glucose environments, which may contribute to diabetic retinopathy [68].

Certain medications prescribed for diabetic patients may already affect mitochondrial fusion or fission in endothelial cells. Empagliflozin, an SGLT-2 inhibitor shown to have clinical benefits in diabetes and heart failure [69,70], may also impact mitochondrial fission dynamics through the suppression of DRP-1, FIS-1, and Mff 1 phosphorylation. In human coronary artery endothelial cells subjected to ischemia reperfusion (IR) injury, empagliflozin administration was able to attenuate vessel barrier dysfunction, cytoskeletal degradation, and eNO synthase suppression caused by IR injury. However, these protective effects were reversed when phosphorylated FIS-1 was reintroduced to subject cells [59]. This indicates that Empagliflozin treats endothelial dysfunction in part through interactions with mitochondrial fission proteins [59]. Metformin is another staple therapeutic agent that has been shown to reduce cardiovascular risk in diabetic patients. Increasing evidence indicates that metformin attenuates mitochondrial fragmentation in part through the AMPK-driven suppression of DRP-1 [71]. In *ApoE*^−/−^ diabetic mice that were administered Metformin, endothelial cells exhibited reduced oxidative stress, improved endothelial function, and smaller atherosclerotic lesions. This result was also re-created with a known mitochondrial fission inhibitor mdivi-1, indicating that mitochondrial fission may be affected by Metformin’s mechanism of action [71] (Figure 3).

## 6. Mitophagy and Cardiovascular Disease

Mitophagy is vital in preserving cellular function. Dysfunction in mitophagy can produce elevated levels of ROS and contribute to intravascular thrombosis, leakage, and inflammation [72]. Hence, protein mediators of mitophagy have gained recent interest for their therapeutic potential in treating CVD.

### 6.1. PINK1 and Parkin

PINK1 or PTEN-induced putative kinase protein and Parkin (RBR E3 ubiquitin-protein ligase) are two proteins heavily involved in mitophagy. PINK1 enters the mitochondria via the translocase complexes of the OMM and IMM, where it is cleaved and dissolved due to its unstable state [73,74,75]. In diseased states, the IMM is depolarized, preventing PINK1 importation to the IMM [76,77]. Hence, PINK1 accumulates on the OMM, recruiting Parkin and other effector proteins, which then ubiquitinate OMM adaptor proteins that aid in the delivery of damaged mitochondria to autophagosomes (Figure 4). The Parkin-PINK1 system has gained interest in CVD over the past decade. Studies in human umbilical vein endothelial cells (HUVECs) demonstrated that exposure to copper oxide nanoparticles, a known oxidative stress promotor, replicates the mechanisms of oxidant stress on endothelial cells [78]. In this model, Parkin-PINK1-mediated mitophagy exerts a protective effect by removing copper oxide-damaged mitochondria [79]. The administration of Mdivi-1, a mitophagy inhibitor, reversed the mitophagy’s protective effect, increasing cell death. Therefore, Parkin-PINK1-mediated mitophagy may help manage oxidative stress in endothelial cells. Additional studies from in vitro models of oxidative stress have allowed for the further study of medications’ effect on mitophagy. For example, in endothelial cells exposed to tert-butyl hydroperoxide (TBHP)-induced oxidative stress, the administration of melatonin reverses EC damage through the upregulation of mitophagy-related proteins Parkin, PINK1, and LC3 [80].

Changes in mitophagy may also contribute to diabetic vascular disease. In studies from in vitro HUVECs exposed to high-glucose and in vivo diabetic rat models, these high-glucose environments are found to decrease PINK1 and Parkin, leading to the accumulation of mitochondrial fragments, ROS overproduction, and endothelial cell apoptosis [81]. Diabetes also leads to advanced glycation end product (AGE) formation through the harmful glycation of the vessel’s components, exacerbating atherosclerosis progression. In vitro treatment of brain endothelial cells with methylglyoxal, the main precursor to AGE, resulted in oxidative stress that demonstrated damaged mitochondria and increased Parkin-mediated mitophagy impairing junctional protein function, increasing endothelial cell permeability [82]. Rivaroxaban and Aspirin combination therapy was also studied for its effects on mitophagy in an in vitro model of diabetic vascular disease [83]. Human coronary artery endothelial cells (HCAECs) incubated in high-glucose environments demonstrated elevated ROS and reduced membrane potential [84]. When treated with Rivaroxaban and Aspirin, these cells demonstrated heightened levels of Parkin and PINK1, restored membrane potential, and reduced ROS generation [84]. These findings suggest that currently used medications may alter mitophagy to provide a therapeutic benefit.

Another example of therapies that affect mitophagy is Scutellarin, a plant extract. Scutellarin has been used to treat cardiovascular disease in herbal medical practice. When treating high-glucose-incubated HUVECs, Scutellarin was shown to upregulate PINK1, Parkin, and MFN; reduce ROS overload; maintain mitochondrial membrane potential; and reduce cell apoptosis [85]. This effect was stifled when the PINK1 gene was knocked out. Hence, the PINK1/Parkin pathway may play a role in Scutellarin’s protective effect against hyperglycemia-induced vascular disease [85].

**Figure 4 pharmaceuticals-18-00112-f004:**
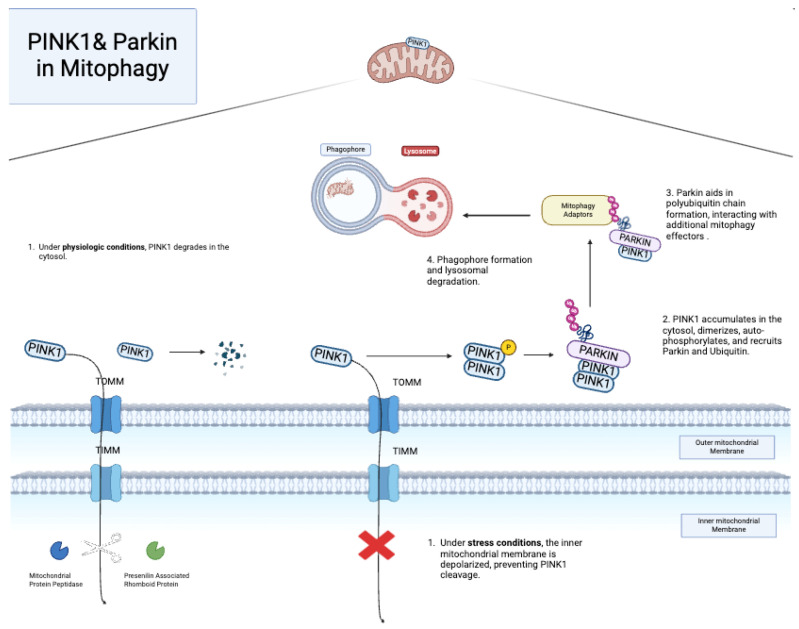
PINK1/Parkin-mediated mitophagy [86].

### 6.2. Other Mitophagy Pathways

Mitophagy can also be achieved through the BNIP3L/Nix pathway. On the OMM, BNIP3L/Nix or Bcl-2-interacting protein-3-like protein inhibits ox-LDL ROS production and NLRP3 inflammasome activation in macrophages through mitophagy [87,88]. The targeted phosphorylation of BNIP3L regulates its interaction with its substrate LC3 for mitochondrial targeting and autophagosome-mediated degradation [89]. BNIP3, or Bcl-2-interacting protein 3, is another mitophagy protein that stabilizes in the OMM during hypoxic states. Upon activation, BNIP3 triggers the mitochondrial depolarization and sequestration of mitochondria into autophagic bodies, resulting in decreased membrane potential, increased ROS production, and decreased energy production [90]. Enhanced BNIP3-induced mitophagy has been implicated in the antioxidant properties of hydrogen sulfide. When cysteine B-synthase, the predominant hydrogen sulfide-producing enzyme, is silenced, endothelial cells demonstrate ER–mitochondria uncoupling, aberrant mitochondrial fission, and subsequent cellular disarray [65].

Another protein, FUNDC-1, mediates mitophagy in hypoxic states with targeted dephosphorylation by upstream proteins [91]. Separately, FUNDC-1 can also indirectly stimulate VEGF2 through interactions with calnexin (CANX) at the endoplasmic reticulum–OMM interface. This interaction promotes angiogenesis through increased VEGFR2 activation [92]. In later stages of hypoxia, FUNDC-1 disassociates from CANX and interacts with fission protein DRP-1 to promote mitochondrial fission and mitophagy [92,93]. FUNDC-1 role in angiogenesis varies on cell type. In endothelial cells, FUNDC-1 specific deletion in EC reduces VEGFR2 expression and inhibits the growth of functional blood vessels [93]. However, in rat aortic VSMC, FUNDC-1 inhibition reduces mitophagy and increases cell proliferation in an Angiotensin (Ang II)-induced proliferation model [91]. Overall, the mechanisms involved in mitophagy and their effects on CVD are intricate and still require further investigation.

### 6.3. Mitochondrial Therapeutics and Their Potential in Non-Cardiovascular Diseases

Recent advances targeting mitochondrial regulation in cardiovascular diseases can also have broader implications in diseases beyond the cardiovascular system. Antioxidants such as Vitamin C, Vitamin E, Coenzyme Q10, alpha-lipoic acid, and Mitoquinol mesylate are being used in Alzheimer’s disease (AD) to mitigate ROS production and subsequent neurodegenerative destruction [94]. Another compound, J-147, which allosterically inhibits mitochondrial ATPase, has been shown to lower oxidative stress and amyloid plaque deposition associated with Alzheimer’s disease [94]. Lastly, metformin, a staple anti-hyperglycemic agent, may also reduce oxidative stress and reduce amyloid plaque deposition as well [94]. This overlap in therapeutic ability highlights the potential of broad implications for mitochondrial therapeutic targeting. Research in multiple other diseases such as chronic kidney disease, chronic obstructive pulmonary disease, and chronic liver disease have also shown that the dysregulation of mitochondrial quality control and mitochondrial homeostasis contributes to disease progression [95,96,97]. Overall, mitochondrial targeting in multiple diseases demonstrates how organelle alterations at the cellular level may result in varied phenotypic changes in different body systems and manifest in a wide array of disease processes. Finally, advances in genomics will also revolutionize our understanding of mitochondrial dysregulation across multiple diseases. Innovations such as Cluster Regulatory Interspaced Short Palindromic Repeats (CRISPR) technology, nanoparticle delivery vectors, and genomic editing have allowed for the recent manipulation of mitochondrial DNA [98]. Tools such as zinc-finger nucleases and transcription-activator-like effector nucleases have succeeded in selectively targeting and eliminating mutated mtDNA in vivo [99,100]. These interventions will allow for a more in-depth analysis of mitochondrial pathways with far-reaching potential beyond the cardiovascular field and the content of the current review.

## 7. Conclusions

Overall, mitochondrial homeostasis is an intricate and dynamic phenomenon that evolved to promote cellular health. However, in diseased states, mitochondrial dysfunction contributes to vessel inflammation and tissue damage. During illness, oxidative stress overwhelms mitochondrial protective mechanisms, resulting in homeostatic disequilibrium. This results in the loss of mitochondrial membrane potential, the release of mtDNA, and the dysregulation of mitochondrial dynamics. Aberrations in these mechanisms have been witnessed in many cardiovascular disease manifestations, and further exploration is crucial to broaden our understanding of them.

## Figures and Tables

**Figure 1 pharmaceuticals-18-00112-f001:**
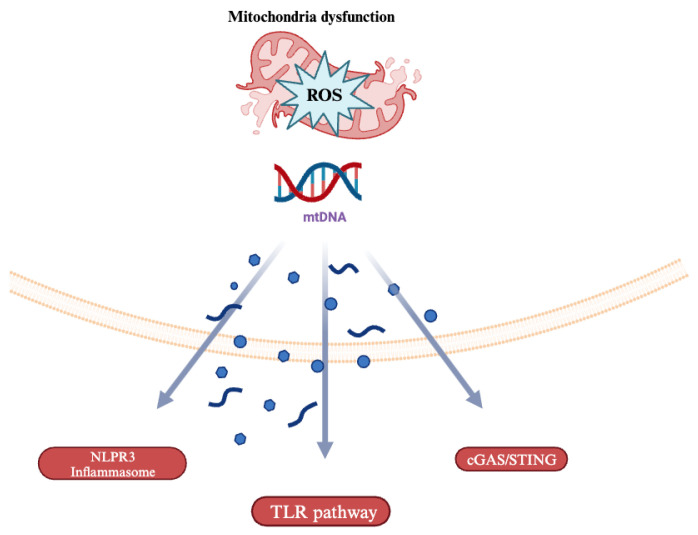
Stress-mediated release of mitochondrial DNA activates multiple pro-inflammatory pathways. Legend: NLRP3: NOD-, LRR-, and pyrin-domain-containing protein 3, TLR: Toll-Like Receptor, cGAS = cyclic GMP-AMP synthase, STING: stimulator of interferon genes.

**Figure 2 pharmaceuticals-18-00112-f002:**
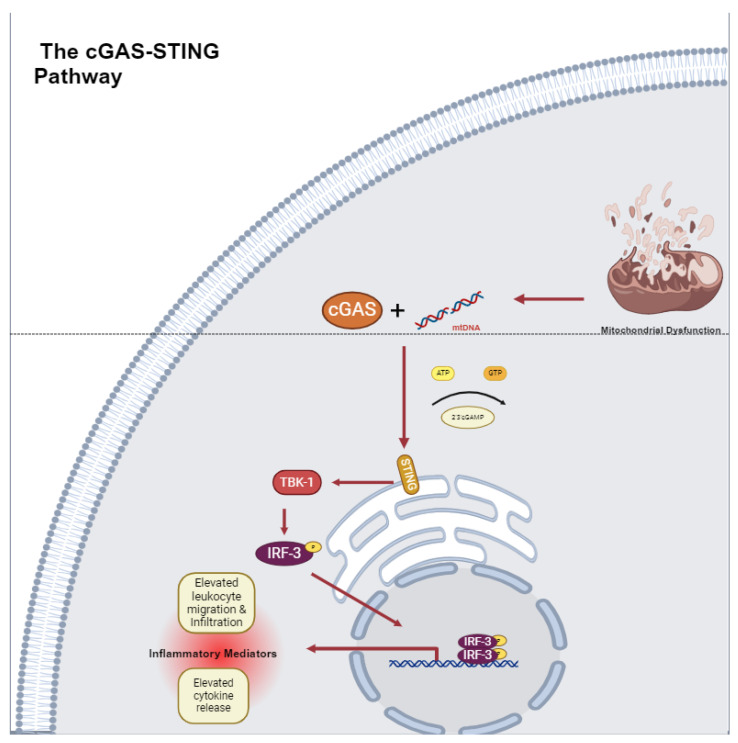
Mitochondrial DNA is released during periods of cellular stress, which is sensed by cGAS. This leads to the activation of STING and TBK-1, which leads to phosphorylation and nuclear translocation of effectors such as IRF-3. These effectors upregulate the transcription of pro-inflammatory cytokines and leukocyte adhesion molecules (i.e., ICAM) [33].

**Figure 3 pharmaceuticals-18-00112-f003:**
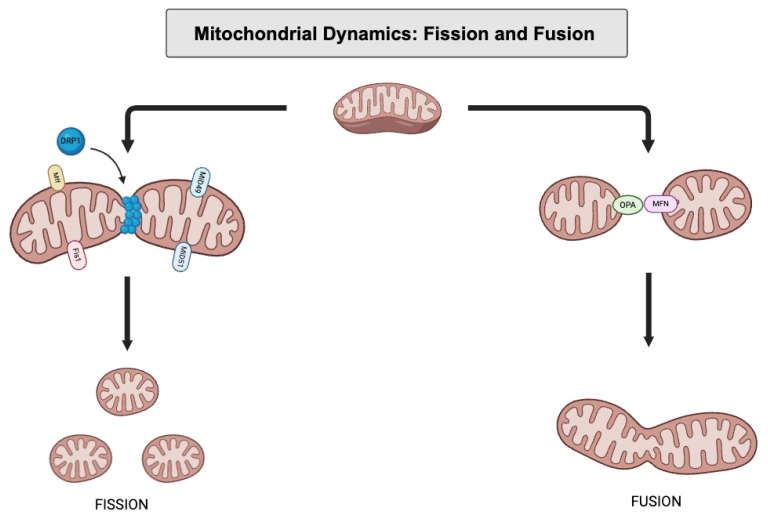
Mitochondrial fission is coordinated by proteins such as DRP-1, which oligomerizes with mitochondrial surface proteins such as FIS-1, Mff, MiD49, and MiD51. Mitochondrial fusion is coordinated by other proteins such as MFN-1, MFN-2, and OPA-1.

## Data Availability

No new data were created during the course of this review article.

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
