# Peer review of "Dysregulation of Mitochondrial Homeostasis in Cardiovascular Diseases"

_pharmaceuticals, 2025, doi:10.3390/ph18010112_

Round 1
Reviewer 1 Report
Comments and Suggestions for Authors
General comment:
The manuscript presented by Patil et al. presents a very detailed review of the effects of mitochondrial dysfunction in cardiovascular diseases at multiple levels. Mitochondrial imbalances in fusion, fission, biogenesis and mitophagy are discussed in the context of vascular diseases. This is a very interesting work and of high interest for the area of research.
Major comments:
i) The title of the article is somewhat counterintuitive since what is discussed in the article are situations of lack of mitochondrial homeostasis.
ii) The authors need to homogenise the designation of abbreviations throughout the manuscript. Many examples are presented below under minor comments.
Minor comments:
i) Page 2 line 70: “enc ephalopathy”; needs correction.
ii) Page 2 line 75: “mechanismfor”; needs correction.
iii) Page 3 line 114: “[21”; needs correction. Same for Page 4 line 133: “[23”.
iv) Page 4 line 147: “TLR 9”; correct to TLR-9.
v) Page 4 lines 144 & 152: the authors use TLR9 instead of TLR-9; they should choose which abbreviation to adopt, but keep it constant.
vi) Page 4 lines 153-154; the authors provide between parenthesis the designation of NLRP3; shouldn’t they do the reverse, have the full name outside parenthesis and the abbreviature between parenthesis?
vii) Page 5 line 185: “an bacteria”; shouldn’t it be a bacteria?
viii) Page 7 line 249: “IRF 3”; sometimes the authors use this, others IRF3; it should be always the same designation.
ix) Page 8 line 266: “OPA 1”; should it be OPA1 or OPA-1 or OPA; too much heterogeneity in designation of abbreviatures.
x) Page 9 lines 301, 302 and 305: the authors use Mfn 2 or Mfn2 or MFN2; again the same issue; lack of homogeneity in referral to abbreviations.
xi) Page 10 line 343: “cultured exposed”; should it be cultures exposed?
xii) Page 10 line 344: “Opa 1”; in previous sections the sdesignation was OPA, now it is Opa; needs homogenization throughout the text.
xiii) Page 10 lines 350 and 353: “Fis1” and “Fis 1”; again heterogenous designations. This issue is present in all the manuscript.
xiv) Page 11 line 363: “Mitochindrial”, “Cooridinated”: words need correction. Also solve the heterogeneity with Mfn 1 versus Mfn2; one has space the other has no space; they both represent proteins.
xv) Page 11 lines 368-369: sentence “Dysfunctional mitochondria can produce elevated levels of ROS and contribute to intravascular thrombosis, leakage, and inflammation, mitophagy is vital to preserving cellular function.”; the way it is written makes no sense.
xvi) Page 11 lines 393-395: “Rivaroxban and Aspirin combination therapy, subjects of the COMPASS trial, was also studied for its effects on mitophagy in an in vitro model of diabetic vascular disease”; what is exactly meant by this sentence?
xvii) Page 11 line 398: “These findings suggest that currently used medications may alter mitophagy to exact their effect”; what is meant by “to exact their effect”? To improve mitophagy? To restore mitophagy?
xviii) Page 12 line 408: The sub-section should be 6.2, not 6.1.
xix) Page 12 line 418: “and subsequent cellular”; something is missing in the sentence.
Author Response
Comment 1: The title of the article is somewhat counterintuitive since what is discussed in the article are situations of lack of mitochondrial homeostasis.
Response 1: Thank you for pointing out this discrepancy and we will modify the title accordingly. We have changed the title to “Dysregulation of Mitochondrial Homeostasis in Cardiovascular Disease”.
Comment 2: The authors need to homogenize the designation of abbreviations throughout the manuscript. Many examples are presented below under minor comments.
Response 2: We thank you and agree with the need to homogenize abbreviation designations throughout the manuscript. We hope the below corrections have addressed this concern.
Minor comments:
Comment 3: Page 2 line 70: “enc ephalopathy”; needs correction.
Response 3: “ Enc ephalopathy” has been corrected to “encephalopathy”. (page 2, line 72)
Comment 4: Page 2 line 75: “mechanismfor”; needs correction.
Response 4: “Mechanismfor” has been corrected to “mechanism for” (page 2, line 77).
Comment 5: Page 3 line 114: “[21”; needs correction. Same for Page 4 line 133: “[23”.
Response 5: The end bracket has been added for both citation [21] and [23].
Comment 6: Page 4 line 147: “TLR 9”; correct to TLR-9.
Response 6: “TLR 9” was corrected to “TLR-9” on Page 4, Line 147.
Comment 7: Page 4 lines 144 & 152: the authors use TLR9 instead of TLR-9; they should choose which abbreviation to adopt but keep it constant.
Response 7: “TLR9” was changed to “TLR-9” on page 4, line 147, line 150, line 151, line 153, line 155.
Comment 8: Page 4 lines 153-154; the authors provide between parenthesis the designation of NLRP3; shouldn’t they do the reverse, have the full name outside parenthesis and the abbreviature between parentheses?
Response 8: On page 4, line 157, parenthesis were removed from the designation of NLRP3 and added to “(NLPR3)”.
Comment 9: Page 5 line 185: “an bacteria”; shouldn’t it be a bacteria?
Response 9: On page 5, line 191 “an bacteria” was changed to “a bacteria”.
Comment 10: Page 7 line 249: “IRF 3”; sometimes the authors use this, others IRF3; should be always the same designation.
Response 10: The phrase IRF-3 was changed on lines 207, 225, 228, 242, 244, 250, 256, 259.
Comment 11: Page 8 line 266: “OPA 1”; should it be OPA1 or OPA-1 or OPA; too much heterogeneity in designation of abbreviatures.
Response 11: All “OPA 1” have been converted to “OPA-1” on lines 274, 276, and 278, and 355.
Comment 12: Page 9 lines 301, 302 and 305: the authors use Mfn 2 or Mfn2 or MFN2; again the same issue; lack of homogeneity in referral to abbreviations.
Response 12: All uses of Mfn have been converted to “MFN” or “MFN-1” or “MFN-2” on lines 274, 304, 315, 316,317, 319, 343, 349, 350, 351, 352, 390, 391, 427, 525.
Comment 13: Page 10 line 343: “cultured exposed”; should it be cultures exposed?
Response 13: On page 10 line 367, “cultured exposed”, the wording was changed to “cultures exposed”.
Comment 14: Page 10 line 344: “Opa 1”; in previous sections the designation was OPA, now it is Opa; needs homogenization throughout the text.
Response 14: All “Opa” abbreviations have been switched to “OPA” or “OPA-1” or “OPA-2” as defined in Response 11.
Comment 15: Page 10 lines 350 and 353: “Fis1” and “Fis 1”; again heterogenous designations. This issue is present in all the manuscript.
Response 15: All mentions of “Fis1” or “Fis 1” have been switched to FIS-1 to establish homogeneity in the manuscript on lines 285, 375, 378, 394, 508.
Comment 16: Page 11 line 363: “Mitochindrial”, “Cooridinated”: words need correction. Also solve the heterogeneity with Mfn 1 versus Mfn2; one has space the other has no space; they both represent proteins.
Response 16: On Page 11 line 393, “mitochondrial” was changed to “mitochondrial” and “cooridinated” was changed to “coordinated”. Additionally, all Mfn proteins are written as “MFN” or “MFN-1” and “MFN-2” to establish homogeneity in the manuscript. This correction was also addressed in Response 12.
Comment 17: Page 11 lines 368-369: sentence “Dysfunctional mitochondria can produce elevated levels of ROS and contribute to intravascular thrombosis, leakage, and inflammation, mitophagy is vital to preserving cellular function.”; the way it is written makes no sense.
Response 17: These sentences have been changed to the following: “Mitophagy is vital to preserving cellular function. Dysfunction in mitophagy can produce elevated levels of ROS and contribute to intravascular thrombosis, leakage, and inflammation”. (lines 398 to 399).
Comment 18: Page 11 lines 393-395: “Rivaroxban and Aspirin combination therapy, subjects of the COMPASS trial, was also studied for its effects on mitophagy in an in vitro model of diabetic vascular disease”; what is exactly meant by this sentence?
Response 18: The COMPASS trial was a large multicenter randomized controlled trial demonstrating that in the treatment of stable cardiovascular disease, dual therapy with Rivaroxaban and Aspirin improved cardiovascular outcomes without significant increase in bleeding risk compared to Rivaroxaban or Aspirin monotherapy. This is the significance of Rivaroxaban and Aspirin combination therapy.
Based on the reviewer’s comments, the authors opted to change the sentence phrasing to “Rivaroxaban and Aspirin combination therapy was also studied for its effects on mitophagy…” on lines 423-424, thus removing the reference to the COMPASS trial to improve sentence clarity.
Comment 19: Page 11 line 398: “These findings suggest that currently used medications may alter mitophagy to exact their effect”; what is meant by “to exact their effect”? To improve mitophagy? To restore mitophagy?
Response 19: Based on the reviewer’s comments, the sentence wording was changed to the following “These findings suggest that currently used medications may alter mitophagy to provide therapeutic benefit” (line 427-428).
Comment 20: Page 12 line 408: The sub-section should be 6.2, not 6.1.
Response 20: On line 451, “6.1 Other Mitophagy Pathways” was changed to “6.2 Other Mitophagy Pathways”.
Comment 21: Page 12 line 418: “and subsequent cellular”; something is missing in the sentence.
Response 21: On line 461, “… and subsequent cellular” was changed to “…and subsequent cellular disarray”.
Reviewer 2 Report
Comments and Suggestions for Authors
The manuscript presents a comprehensive and insightful review of the role of mitochondrial homeostasis in cardiovascular diseases (CVD). It effectively outlines the critical processes of mitochondrial dynamics, oxidative stress, and mitophagy and discusses their implications in CVD pathogenesis and potential therapeutic strategies. The topic is timely, given the growing interest in targeting mitochondrial dysfunction for therapeutic intervention in cardiovascular diseases.
The manuscript addresses an important topic in cardiovascular research and is well-written. I will recommend the acceptance after minor revisions (listed below).
The manuscript discusses potential interventions targeting mitochondrial pathways but lacks depth in exploring emerging therapeutic strategies. The inclusion of a paragraph summarising recent innovations, such as CRISPR-based tools for mitochondrial genome editing or advanced nanoparticle delivery systems, could enhance the discussion of future directions.
The manuscript focuses exclusively on mitochondrial dysfunction in CVD. The addition of a brief comparative discussion with other diseases, such as Alzheimer's disease or diabetes, could provide valuable context. For instance, common mechanisms such as mitochondrial DNA damage, oxidative stress, and altered energy production are also central to these diseases. Highlighting these parallels would underline the broader relevance of mitochondrial dysfunction as a therapeutic target. To enhance the discussion on therapeutic strategies, the authors could refer to this article: DOI: 10.3390/life11050386. This study illustrates the role of mitochondrial dysfunction in another chronic disease, emphasizing oxidative stress and metabolic alterations. The proteomics and therapeutic approaches discussed in this work may offer insights applicable to CVD, particularly regarding strategies to mitigate oxidative damage and restore mitochondrial homeostasis.
Author Response
Comment 1: The manuscript presents a comprehensive and insightful review of the role of mitochondrial homeostasis in cardiovascular diseases (CVD). It effectively outlines the critical processes of mitochondrial dynamics, oxidative stress, and mitophagy and discusses their implications in CVD pathogenesis and potential therapeutic strategies. The topic is timely, given the growing interest in targeting mitochondrial dysfunction for therapeutic intervention in cardiovascular diseases.
The manuscript addresses an important topic in cardiovascular research and is well-written. I will recommend the acceptance after minor revisions (listed below).
The manuscript discusses potential interventions targeting mitochondrial pathways but lacks depth in exploring emerging therapeutic strategies. The inclusion of a paragraph summarising recent innovations, such as CRISPR-based tools for mitochondrial genome editing or advanced nanoparticle delivery systems, could enhance the discussion of future directions.
The manuscript focuses exclusively on mitochondrial dysfunction in CVD. The addition of a brief comparative discussion with other diseases, such as Alzheimer's disease or diabetes, could provide valuable context. For instance, common mechanisms such as mitochondrial DNA damage, oxidative stress, and altered energy production are also central to these diseases. Highlighting these parallels would underline the broader relevance of mitochondrial dysfunction as a therapeutic target.
To enhance the discussion on therapeutic strategies, the authors could refer to this article: DOI: 10.3390/life11050386. This study illustrates the role of mitochondrial dysfunction in another chronic disease, emphasizing oxidative stress and metabolic alterations. The proteomics and therapeutic approaches discussed in this work may offer insights applicable to CVD, particularly regarding strategies to mitigate oxidative damage and restore mitochondrial homeostasis.
Response 1:
The article DOI: 10.3390/life11050386. was briefly referenced to discuss mitochondrial targeting in Alzheimer’s disease.
The following paragraph was added to the manuscript from lines 475 to 497:
“6.3. Mitochondrial Therapeutics and their potential in non-cardiovascular diseases:
Recent advance targeting mitochondrial regulation in cardiovascular diseases can also have broader implications in diseases beyond the cardiovascular system. Antioxidants such as Vitamin C, Vitamin E, Coenzyme Q10, alpha-lipoic acid, and Mitoquinol mesylate are being used in Alzheimer’s Disease (AD) to mitigate ROS production and subsequent neurodegenerative destruction [95]. Another compound, J-147, which allosterically inhibits mitochondrial ATPase, has been shown to lower oxidative stress and amyloid plaque deposition associated with Alzheimer’s disease [95]. Lastly, metformin, a staple anti-hyperglycemic agent, may also reduce oxidative stress and reduce amyloid plaque deposition as well [95]. This overlap in therapeutic ability highlights the potential of broad implications for mitochondrial therapeutic targeting. Research in multiple other diseases such as chronic kidney disease, chronic obstructive pulmonary disease, and chronic liver disease have also shown that dysregulation of mitochondrial quality control and mitochondrial homeostasis contributes to disease progression [96, 97, 98]. Overall, mitochondrial targeting in multiple diseases demonstrates how organelle alterations at the cellular level may result in varied phenotypic changes in different body systems and manifesting in a wide array of disease processes. Finally, advances in genomics will also revolutionize understanding of mitochondrial dysregulation across multiple diseases. Innovations such as Cluster Regulatory Interspaced Short Palindromic Repeats (CRISPR) technology, nanoparticle delivery vectors, and genomic editing have allowed for the recent manipulation of mitochondrial DNA [99]. Tools such as zinc-finger nucleases and transcription activator-like effector nucleases have succeeded in selectively targeting and eliminating mutated mtDNA in vivo [100, 101]. These interventions will allow for a more in-depth analysis of mitochondrial pathways with far reaching potential beyond the cardiovascular field and the content of the current review”.